# Camu-Camu Fruit Extract Inhibits Oxidative Stress and Inflammatory Responses by Regulating NFAT and Nrf2 Signaling Pathways in High Glucose-Induced Human Keratinocytes

**DOI:** 10.3390/molecules26113174

**Published:** 2021-05-26

**Authors:** Nhung Quynh Do, Shengdao Zheng, Bom Park, Quynh T. N. Nguyen, Bo-Ram Choi, Minzhe Fang, Minseon Kim, Jeehaeng Jeong, Junhui Choi, Su-Jin Yang, Tae-Hoo Yi

**Affiliations:** 1Graduate School of Biotechnology, Kyung Hee University, 1732 Deogyeong-Daero, Giheung-gu, Yongin-si 17104, Gyeonggi-do, Korea; quynhnhung96@khu.ac.kr (N.Q.D.); sdjeong0719@khu.ac.kr (S.Z.); viesna91@khu.ac.kr (B.P.); quynhnguyen@khu.ac.kr (Q.T.N.N.); mincheolbang1030@gmail.com (M.F.); dbs03067@khu.ac.kr (M.K.); junhi4703@khu.ac.kr (J.C.); 2Department of Herbal Crop Research, National Institute of Horticultural and Herbal Science, RDA, Eumseong 27709, Korea; bmcbr@korea.kr; 3Snow White Factory Co., Ltd., 807 Nonhyeonro, Gangnam-gu, Seoul 06032, Korea; khbubu@hanmail.net; 4Gu Star Co., Ltd., 7/F, Cheongho B/D, 19, Eonju-ro 148-gil, Gangnam-gu, Seoul 06054, Korea; sj@gustar.co.kr

**Keywords:** camu-camu fruit, oxidative stress, inflammation, high glucose, keratinocytes, NFAT, Nrf2

## Abstract

*Myrciaria dubia* (HBK) McVaugh (camu-camu) belongs to the family Myrtaceae. Although camu-camu has received a great deal of attention for its potential pharmacological activities, there is little information on the anti-oxidative stress and anti-inflammatory effects of camu-camu fruit in skin diseases. In the present study, we investigated the preventative effect of 70% ethanol camu-camu fruit extract against high glucose-induced human keratinocytes. High glucose-induced overproduction of reactive oxygen species (ROS) was inhibited by camu-camu fruit treatment. In response to ROS reduction, camu-camu fruit modulated the mitogen-activated protein kinases (MAPK)/activator protein-1 (AP-1), nuclear factor kappa-light-chain-enhancer of activated B cells (NF-κB), and nuclear factor of activated T cells (NFAT) signaling pathways related to inflammation by downregulating the expression of proinflammatory cytokines and chemokines. Furthermore, camu-camu fruit treatment activated the expression of nuclear factor E2-related factor 2 (Nrf2) and subsequently increased the NAD(P)H:quinone oxidoreductase1 (NQO1) expression to protect keratinocytes against high-glucose-induced oxidative stress. These results indicate that camu-camu fruit is a promising material for preventing oxidative stress and skin inflammation induced by high glucose level.

## 1. Introduction

*Myrciaria dubia* (HBK) McVaugh (camu-camu) belongs to the Myrtaceae family that is native to the Amazon rainforest. Its fruit is well-studied for its antioxidative activities both in vitro and in vivo through the decrease of reactive oxygen species (ROS) production [1,2,3]. Camu-camu exhibits pharmacological effects on neuroprotection, antihypertension, antimicrobial, anti-proliferation, and anti-genotoxicity activities [4,5,6,7]. These activities are due to the effects of the vitamin C content, promising polyphenols such as gallic acid, quercetin, and ellagic acid, and proanthocyanins such as cyanidin-3-glycoside. These active compounds exhibit various biological properties which are listed in Table 1. Although camu-camu has received much attention for its potential pharmacological activities, there is little information on the effects of underlying anti-inflammatory mechanisms of fruit extracts in skin diseases.

Inflammation and oxidative stress represent biological responses of an organism to chemicals, infectious stimuli, and injuries. However, prolonged oxidative stress results in imbalance between production and elimination of reactive oxygen species (ROS) by the antioxidant defense system [8]. This action can lead to chronic inflammation and cause acute and chronic diseases like psoriasis and atopic dermatitis [9,10]. Among stress-induced factors, high glucose was characterized as a proinflammatory condition. In the human body, plasma glucose is maintained at 3.9–5.6 mM, and a level higher than 6.9 mM is considered hyperglycemia which can potentially lead to an acute and/or chronic, life-threatening condition [11,12]. High glucose significantly triggers oxidative stress by increasing the activity of the mitochondrial respiratory chain and promoting the production of oxygen free radicals (e.g., ROS) [13]. Dysregulation of glucose homeostasis and elevated glucose level contribute to impaired wound healing by upregulating inflammatory markers [14]. Both are involved in overstimulating the immune responses to result in development of chronic human diseases [15]. High glucose is studied for its effects on impaired wound healing, but its underlying mechanisms of high-glucose-induced skin inflammation have not been elucidated.

Furthermore, nuclear factor of activated T cells (NFAT) is a family of calcineurin-dependent transcription factors that first was identified in T cells and has been found broadly in other organs such as skin tissues where it plays a crucial role in immune response [16,17]. When the transcription factor NFAT is inactivated, its phosphorylated form is localized in the cytoplasm. Exposure to inflammatory stimuli activates the NFAT transcription factor. The activation of NFAT results in dephosphorylation of NFAT proteins and translocation of NFAT to the nucleus that are related to the NF-κB family of transcription factors. Once activated, NFAT along with AP-1 regulate the transcription of interleukin-6 (IL-6), IL-8, and COX-2 which are implicated in the inflammatory response [18,19,20].

The Nrf2 signaling pathway plays a pivotal role in the natural antioxidant defense system against oxidative stress. Transcription factor Nrf2, which regulates the oxidative stress tolerance and lifespan, translocates from the cytosol to the nucleus to activate the ARE-dependent gene and increase antioxidant enzymes (HO-1, NQO1) [21]. Thus, increased expression of Nrf2 and antioxidant enzymes such as HO-1 and NQO1 can protect skin against oxidative damage [22,23,24]. However, prolonged ROS overproduction causes damage of cellular structure and function, resulting in dysfunction and cell death. This injury impairs the Nrf2/ARE pathway so that it no longer protects cells against excessive ROS generation [21].

Thus, in this study, we aimed to assess the inhibitory mechanisms of the camu-camu fruit extract on skin inflammatory responses by suppressing oxidative stress. Our results indicated that the camu-camu fruit could control the expression of inflammatory mediators by upregulating Nrf2 activation and downregulating the NFAT and NF-κB, AP-1, MAPK signaling pathways by inhibiting ROS production in high glucose-induced immortal keratinocytes.

## 2. Results

### 2.1. Analysis of Polyphenolic Compounds in the Camu-Camu Fruit Extract

HPLC analysis was performed to determine the contents of ellagic acid and quercetin in the camu-camu fruit. As shown in the chromatogram (Figure 1), ellagic acid and quercetin were confirmed as active compounds at 25.28 min and 36.55 min. The contents of ellagic acid and quercetin in 10 mg/mL of the camu-camu fruit extracted with 70% ethanol plus 1% formic acid were 1.412% and 0.09 %, respectively.

### 2.2. Antioxidative Activities of the Camu-Camu Fruit Extract

The free radical-scavenging activities of camu-camu and ascorbic acid were determined by DPPH inhibition. As shown in Figure 2A, the DPPH radical-scavenging activity of both camu-camu and ascorbic acid was increased considerably in a dose-dependent manner. Notably, the scavenging activity of camu-camu was 95.7% at 250 μg/mL, with the IC_50_ value of 17.95 μg/mL.

To investigate effects of the camu-camu fruit extract on intracellular ROS production in high-glucose-stimulated oxidative stress, HaCaT cells were subjected to 15 mM D-glucose for 6 h and detected by the DCF-DA fluorescent dye. ROS generation was increased by treatment with high glucose (up to 138%); however, camu-camu fruit extract and dexamethasone treatment successfully inhibited this ROS overproduction (Figure 2B,C). HaCaT cells pretreated with 10 µM dexamethasone exhibited a 25.7% decrease in ROS production, while supplementation with the camu-camu fruit reduced ROS production by 40% at 1 μg/mL and 80% at 10 μg/mL compared with the high glucose-treated cells.

### 2.3. Effects of the Camu-Camu Fruit Extract on Cell Viability and mRNA Expression of Proinflammatory Cytokines and Chemokines in High Glucose-Induced HaCaT Cells

The MTT assay was performed to determine the cytotoxic effect of the camu-camu fruit extract on high glucose-stimulated HaCaT cells. The cells exposed to high glucose did not exhibit any significant decrease. The camu-camu fruit extract was not cytotoxic to cells at the applied concentrations (1, 10 μg/mL).

Proinflammatory expression induced by high glucose was analyzed in HaCaT cells in the presence of the camu-camu fruit extract (1, 10 µg/mL); dexamethasone was used as a positive control. Figure 3B–F shows that the expression of MDC, RANTES, and TARC was upregulated in high-glucose-induced HaCaT cells, up to 170.1%, 426.8%, and 180.5%, respectively, whereas IL-8 exhibited a moderate increase to 137.9% compared with that of the untreated cells. These increases were strongly inhibited by treatment with dexamethasone and the camu-camu fruit extract. The camu-camu fruit extract exhibited equivalent inhibition in the expression of TARC and RANTES compared to dexamethasone and greater inhibition than treatment with dexamethasone in the expression of MDC. In particular, supplementation with 10 µg/mL camu-camu fruit extract counteracted the upregulated expression of MDC by 65.1%, while 10 µM dexamethasone did not show any changes.

### 2.4. Inhibitory Effects of the Camu-Camu Fruit Extract on MAPKs and AP-1 Activation in High-Glucose-Induced HaCaT Cells

The MAPK signaling pathway plays a pivotal role in the regulation of AP-1 as well as in inflammatory responses. High glucose was indicated to activate the MAPK signaling pathway via overproduction of oxidative stress. As shown in Figure 4A–D, high-glucose-induced HaCaT cells upregulated p-ERK, p-JNK, and p-p38 to 130.7%, 140.3%, and 237.5%, respectively. In contrast, addition of the camu-camu fruit extract to the treatment downregulated phosphorylation of MAP kinases signaling in a dose-dependent manner. Treatment with 10 µg/mL camu-camu fruit extract was more effective than the positive controls, dexamethasone and cyclosporin A. In the controls, 10 µg/mL camu-camu fruit extract suppressed p-p38 to 56.4%, while 10 µM dexamethasone decreased p-p38 by 23.3% and cyclosporin A inhibited the expression of p-p38 by 45.5% as compared with that of high glucose-stimulated cells. In general, cyclosporin A exhibited better inhibition of MAPK than dexamethasone. This manifested as decrease of p-JNK and p-p38 protein expression.

AP-1 is a transcription factor comprised of c-Jun and c-Fos subunits transduced by the MAP kinase at both transcriptional and post-translational levels. To determine whether the camu-camu fruit extract inhibited high glucose-induced AP-1, we measured protein expression of c-Jun and its phosphorylated form. As shown in Figure 4E,F high glucose upregulated p-c-Jun protein expression to 487.4% compared to that of the untreated cells. This upregulation was attenuated under camu-camu fruit treatment but neither by dexamethasone nor cyclosporin A. Treatment of cells with 10 µg/mL camu-camu fruit extract successfully inhibited the expression of p-c-Jun by 44.5% compared to that of high-glucose-induced HaCaT cells.

### 2.5. Inhibitory Effects of the Camu-Camu Fruit Extract on NF-κB in High Glucose-Induced HaCaT Cells

Transcription factor NF-κB serves as an essential mediator in the inflammatory response by modulating the expression of proinflammatory genes including cytokines and chemokines, as well as cell survival [38]. Inducible degradation of the inhibitor IkB kinase is a primary mechanism of NF-κB activation [39]. Phosphorylation of IkBα in high-glucose-induced HaCaT cells coordinates with NF-κB activation. Thus, we investigated whether the camu-camu fruit mediates inhibition of phosphorylation of IkBα and NF-κB protein expression. High glucose adequately triggered phosphorylation of IkBα causing NF-κB activation to 231.7% and 135.2% in p-IkBα and NF-κB, respectively (Figure 5). As expected, addition of 1 µg/mL camu-camu fruit extract inhibited high glucose-induced p-IkBα and NF-κB by 31.5% and 30.1%, respectively, as compared to that of high glucose-induced cells. On the other hand, dexamethasone and cyclosporin A decreased NF-κB protein expression by 32.5% and 15.7% and p-IkBα protein expression by 51.3% and 25.6%, respectively.

### 2.6. Inhibitory Effects of the Camu-Camu Fruit Extract on p-NFATc1 in High Glucose-Induced HaCaT Cells

To determine whether NFATc1 expression is affected in high glucose-induced HaCaT cells, the cells were triggered with 15 mM D-glucose for 4 h. Referred to Figure 6 high glucose promoted NFATc1 dephosphorylation to 89.8% compared to that of the non-induced cells, but this effect was inhibited by camu-camu fruit treatment. In particular, preincubation with 10 µg/mL camu-camu fruit increased phosphorylated NFATc1 by 158.1% compared to that of high glucose-induced HaCaT cells. There were no increases in the level of phosphorylated NFATc1 with dexamethasone or cyclosporin A. In addition, activated COX-2 signaling is regulated with the increase of NFAT expression [40]. COX-2 expression was activated by induction of high glucose up to 148.8%, but dexamethasone, cyclosporin A, and camu-camu fruit treatment reversed this tendency by 56.2%, 38.6%, and 23.4%, respectively.

### 2.7. Inhibitory Effects of the Camu-Camu Fruit Extract in Nrf2 in High Glucose-Induced HaCaT Cells

Downregulation of ROS levels is mediated by the Nrf2/ARE signaling pathway. To investigate whether the camu-camu fruit extract inhibits the production of ROS, we measured the expression of the transcription factor Nrf2 and Nrf2-related antioxidant proteins in high-glucose-induced immortal keratinocyte HaCaT cells. As shown in Figure 7, Nrf2 was activated by high-glucose-induced oxidative stress. Treatment with the camu-camu fruit extract exacerbated the expression of the Nrf2 protein by 59.8% compared with that of high-glucose-induced HaCaT cells. Activation of transcription factor Nrf2 strengthened the antioxidant defense system by enhancing the expression of NQO1 to 214.4%, compared with that of high glucose-induced HaCaT cells. Dexamethasone and cyclosporin A had no or little effect on facilitating Nrf2 and the antioxidant defense protein.

## 3. Discussion

The camu-camu fruit is a tropical fruit native to the Amazon region and is one of the few Amazon fruits that have been explored for commercial purposes [41]. The phenolic components of the camu-camu fruit include ellagic acid and quercetin which contribute to the pharmacological properties. In this study, we aimed to study the underlying mechanisms of the inhibitory effect of the camu-camu fruit extract on skin inflammatory responses in high glucose-induced keratinocytes (Figure 8).

The results showed that the camu-camu fruit dose-dependently inhibited mRNA expression of proinflammatory cytokine IL-8 and chemokines MDC, RANTES, and TARC in high-glucose-induced HaCaT cells. In most cases, the camu-camu fruit and dexamethasone comparatively exhibited strong suppression of the expression of cytokines and chemokines. However, the camu-camu fruit blocked the expression of MDC by 72.3% at the concentration of 10 µg/mL compared with that with 10 µM dexamethasone treatment. Epidermal keratinocytes are important providers of neutrophil and granulocyte chemotactic cytokines including interleukin (IL)-8, which then migrate to injured sites. Other chemokines such as TARC/CCL17, MDC/CCL22, and RANTES/CCL5 are typical Th2 cell-secreted chemokines predominantly expressed by keratinocytes involved in atopic dermatitis. Thus, we assume that the camu-camu fruit has a potential in inhibiting the expression of inflammatory mediators to regulate the anti-inflammatory reaction.

It was well-demonstrated that a high-glucose condition upregulates the activation of NF-κB and AP-1 as well as its upstream signaling p38/MAPK pathway in the presence of high levels of ROS [4,42,43]. Activation of NF-κB and AP-1 translocation to the nucleus result in transcription of genes such as IL-8, TARC, and RANTES [44]. It has been indicated that ellagic acid treatment reduces the increase of ROS generation, and ERK activation in high glucose-induced HAEC cells [32]. Similar to that study, our results indicated that high-glucose treatment upregulated the expression of NF-κB and MAPK/AP-1 in human keratinocyte cells. The camu-camu fruit effectively downregulated phosphorylation of AP-1 and MAPKs compared with the effects of dexamethasone and cyclosporin A. Hence, the camu-camu fruit exhibited better anti-inflammatory effects by controlling inflammatory signaling pathways compared to positive controls in high-glucose-induced HaCaT cells.

We also examined expression of the transcription factor NFAT, which has been reported to be stimulated by the high-glucose condition by activating calcium/calcineurin pathways [45,46]. Once activated, NFAT dephosphorylates and transports to the nucleus to transcribe the expression of inflammatory mediators such as IL-8 and COX-2 [19,40]. Cyclosporin A is a calcineurin inhibitor with immunosuppressive actions and is an effective treatment for many inflammatory diseases including skin diseases such as atopic dermatitis [47,48]. However, its widespread and longer-term use is limited because of serious side effects that include nephrotoxicity and hypertension. Therefore, substitute products for skin treatment are needed, and there is a growing demand for incorporation of natural products in functional foods as they might have fewer side effects than chemical-based products. Interestingly, we found that the camu-camu fruit blocked NFATc1 dephosphorylation stronger than cyclosporin A, increasing phosphorylated NFATc1 to 165.4% compared with that of cyclosporin A-treated HaCaT cells (Figure 6A,B). Thereby, the camu-camu fruit can be used to attenuate the transcription of inflammatory factors IL-8 and COX-2.

Oxidative stress reflects an imbalance between ROS generation and elimination. Activation of the Nrf2 transcription factor was demonstrated to balance ROS generation by regulating the expression of numerous antioxidant-related genes. But high-glucose treatment significantly decreased the expression of Nrf2 in a time-dependent manner [4]. Previous studies have showed that ellagic acid reduced oxidative stress and insulin resistance in diabetic female rats by increasing the Nrf2 protein expression [25,49]. In the present study, even at low concentrations (1 and 10 µg/mL), the camu-camu fruit extract exhibited better preventative effects against high glucose-induced ROS generation by regulating the Nrf2 pathway than that of dexamethasone and cyclosporin A (Figure 7). Particularly, 10 µg/mL camu-camu fruit increased the Nrf2 and NQO1 expression 2.14- and 3.9-fold, respectively, compared to the increase by dexamethasone. In comparison to cyclosporin A treatment, camu-camu fruit produced 1.5-fold and 1.7-fold increases, respectively. These results suggest that the camu-camu fruit extract exerts great antioxidant activity through upregulation of the Nrf2 signaling pathway.

## 4. Materials and Methods

### 4.1. Sample Preparation

To prepare the camu-camu extract, dried camu-camu fruit (Mountain Rose Herbs Co., Eugene, OR, USA) was extracted three times with 70% ethanol plus 1% formic acid in a SHO-1D digital orbital shaker (Daihan, Korea) at 24–25 °C. After 24 h incubation, the camu-camu extract was filtered and condensed using a rotary vacuum evaporator at 40 °C (EYELA WORLD-Tokyo Rikakikai Co., Ltd., Tokyo, Japan). The final yield was 19.96 % (*w*/*v*).

### 4.2. High-Performance Liquid Chromatography Analysis

An UltiMate 3000 LC high-performance liquid chromatography (HPLC) system (Thermo Fisher Scientific and Voorhees Scientific Inc., Sunnyvale, CA, USA) with a 4.6 × 250 mm^2^, 5-μm Hypersil GOLD C18 column (Thermo Fisher Scientific, Waltham, MA, USA) was used to detect polyphenolic compounds in the camu-camu extract. Briefly, 10 μL of the extract were injected in 1 mg/mL 50% methanol at a flow rate of 1 mL/min and a detection wavelength of 280 nm. Ellagic acid and quercetin standards (Sigma-Aldrich, St. Louis, MO, USA) were prepared in 50% methanol in a range of 5 μg/mL to 1000 μg/mL. For ellagic acid, the detection wavelength of the camu-camu fruit was set at 254 nm, and that of quercetin was 360 nm. The working sample of the camu-camu fruit extract was prepared in 100% methanol at 10 mg/mL. The flow rate was 1 mL/min, and the injection volume was 50 μL.

### 4.3. DPPH Scavenging Activity

The scavenging activity of camu-camu was assessed by the 1,1-diphenyl-2-picrylhydrazyl (DPPH) assay with a final concentration of 0.2 mM. Both camu-camu and the positive control, ascorbic acid, which was purchased from Sigma-Aldrich, St, Louis, MO, USA, were used at the concentrations of 1, 10, 50, 100, and 250 μg/mL and incubated with DPPH for 30 min at 37 °C. Absorbance was detected at 520 nm using a FilterMax F5 microplate reader (Molecular Devices, San Francisco, CA, USA). The percentage of DPPH radical inhibition was determined using the following formula:(1)$$\%\;\mathrm{of}\;\mathrm{DPPH}\;\mathrm{radical}\;\mathrm{inhibition}=\frac{\mathrm{ODo}\;-\mathrm{ODx}}{\mathrm{ODo}}\times 100$$

### 4.4. Cell Culture, Camu-Camu Treatment, and Stimulation

Human immortal keratinocyte HaCaT cells were cultured and incubated in the Dulbecco’s modified Eagle medium (DMEM medium, normal glucose, 4.5 mM) (Gibco-BRL, Grand Island, NY, USA) supplemented with 10% heat-inactivated fetal bovine serum and 1% penicillin−streptomycin. They were maintained in an incubator with the 5% CO_2_ atmosphere at 37 °C. HaCaT cells were seeded at a density of 3.0 × 10^5^ cells/mL. Inflammation was induced in HaCaT cells with D-glucose (Daejung Chemicals & Metals Co., Ltd., Siheung, Korea). HaCaT cells were incubated with the camu-camu fruit extract for 1 h before being stimulated with 15 mM D-glucose for 6 h.

### 4.5. Cell Viability Assay

Cell viability was measured by solubilizing purple formazan crystals, after which the difference in color was determined spectrophotometrically. Briefly, supernatants were discarded after treatment and the MTT reagent was added to the final concentration of 0.1 mg/mL. After 2 h incubation at 37 °C, all supernatants were removed and 100 μL dimethyl sulfoxide was added to dissolve the formazan crystals. The optical density at 570 nm was recorded using a FilterMax F5 microplate reader.

### 4.6. Measurement of ROS Generation

After 6 h of treatment with high glucose, the cells were stained with 30 μM 2′7′-dichloro-fluorescein diacetate (DCF-DA) (Sigma-Aldrich) for 30 min at 37 °C in a CO_2_ incubator. Subsequently, the cells were washed thrice with ice-cold phosphate buffer saline (PBS) and suspended in PBS. The cells were then detected at an excitation wavelength of 485 nm and an emission of 535 nm by flowcytometry (FACSCalibur™; Becton-Dickinson, San Jose, CA, USA). Fluorescence intensity was gated with 10,000 events and the mean of fluorescence intensity was used to represent the amount of ROS generation.

### 4.7. Reverse Transcription-Polymerase Chain Reaction

Total cellular RNA was extracted from HaCaT cells using TRIzol (Invitrogen Co., Grand Island, USA) and the subsequent steps were described in a previous study [50]. Briefly, the concentration of RNA samples was quantified and the reverse transcriptase reaction was carried out for 60 min at 42 °C, followed by 5 min at 94 °C. A PCR premix (Bioneer, Korea) and a Veriti Thermal Cycler (Applied Biosystems, Foster City, CA, USA) were used for polymerase chain reaction (PCR) amplification. Primers for IL-8, MDC, RANTES, TARC, GAPDH were provided in Appendix A. PCR products were separated by 2.0% agarose gel electrophoresis and visualized with ethidium bromide staining under UV illumination. The density of PCR products was normalized to that of glyceraldehyde 3-phosphate dehydrogenase (GAPDH).

### 4.8. Western Blot Analysis

After treatment with the camu-camu fruit and exposure to high glucose, the protein levels in total cell lysates were calibrated using the Bradford reagent (Bio-Rad, Hercules, CA, USA) and bovine serum albumin (BSA) as the standard. Equal quantities of the total protein were separated by sodium dodecyl sulfate polyacrylamide gel electrophoresis (SDS-PAGE) and then transferred to a PVDF membrane (Bio-Rad). Transfer membranes were blocked in 5% skim milk or BSA prepared in TBST. The membrane was immersed in an appropriate primary antibody (Cell Science, Canton, MA and Santa Cruz Biotechnology Inc., Santa Cruz, CA, USA) and shaken overnight at 4 °C. After washing and incubating with a secondary antibody for 1 h at room temperature, the membrane was stained with the West-Q Pico ECL solution (GenDepot, Barker, TX, USA), and protein levels were detected using electrochemiluminescence (Fujifilm, LAS-4000, Tokyo, Japan) and quantified by ImageMasterTM 17 2D Elite software version 3.1 (Amersham Pharmacia Biotech, Piscataway, NJ, USA).

### 4.9. Statistical Analysis

The results were analyzed using the Prism 5 statistical analysis software (GraphPad). The data were expressed as the means ± standard deviation (SD). Statistical comparisons between treatments were performed using one-way analysis of variance, followed by Duncan’s test. Student’s *t*-tests were used to compare individual treatments to the control; *p* < 0.05 was considered statistically significant.

## 5. Conclusions

In conclusion, this study provides scientific evidence of the mechanisms underlying oxidative stress reduction and anti-inflammatory effects of the camu-camu fruit on high glucose-induced skin damage. Our results demonstrated that the camu-camu fruit modulates the NF-κB/AP-1, MAPK, and NFAT signaling pathways related to inflammation by downregulating the expression of proinflammatory cytokines and chemokines. More importantly, Nrf2 was used as a pharmacological target to upregulate the antioxidant defense enzyme to protect HaCaT cells against high-glucose-induced oxidative stress. Using the camu-camu fruit instead of cyclosporin A therapy is a potential option for treatment of skin diseases. Therefore, the camu-camu fruit is a promising material for prevention and treatment of high glucose-induced skin inflammation and is expected to be applicable as functional foods and medicines as a dual-functional material for improving human skin disorders.

## Figures and Tables

**Figure 1 molecules-26-03174-f001:**
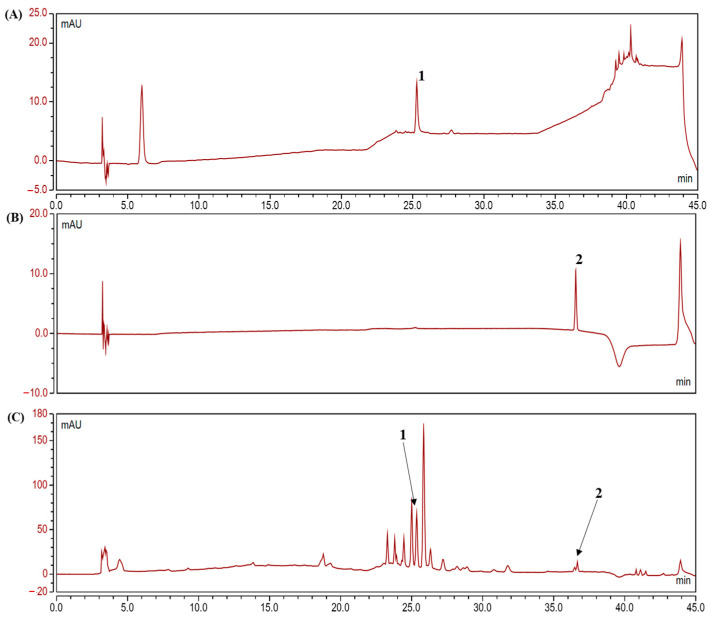
The HPLC (high-performance liquid chromatography) results of ellagic acid (peak 1) and quercetin standards (peak 2) (**A**,**B**) and the contents of ellagic acid and quercetin in the camu-camu fruit extract (**C**).

**Figure 2 molecules-26-03174-f002:**
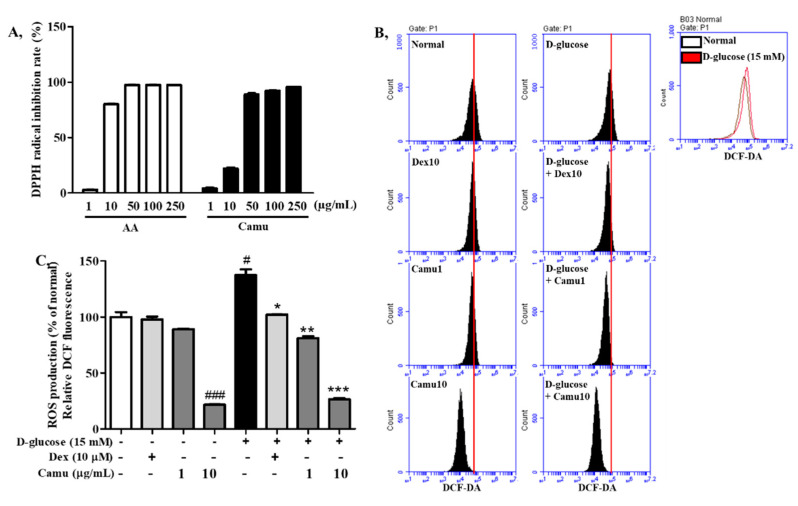
Antioxidative activities of the camu-camu fruit extract. DPPH radical-scavenging activity of the camu-camu fruit extract (**A**). Levels of ROS production in the HaCaT cells treated with 15 mM D-glucose for 6 h (**B**,**C**). The cells were stained with DCF-DA for 30 min; then, the number of cells was plotted versus the dichlorofluorescein detected by the FL-2 channel. All the data are shown as the means ± SD of three independent experiments. # Significant differences between the untreated group and the high glucose-induced group. (# *p* < 0.05, ### *p* < 0.001). * Significant differences between the high glucose-induced group and the other groups (* *p* < 0.05; ** *p* < 0.01; *** *p* < 0.001). Dex, dexamethasone.

**Figure 3 molecules-26-03174-f003:**
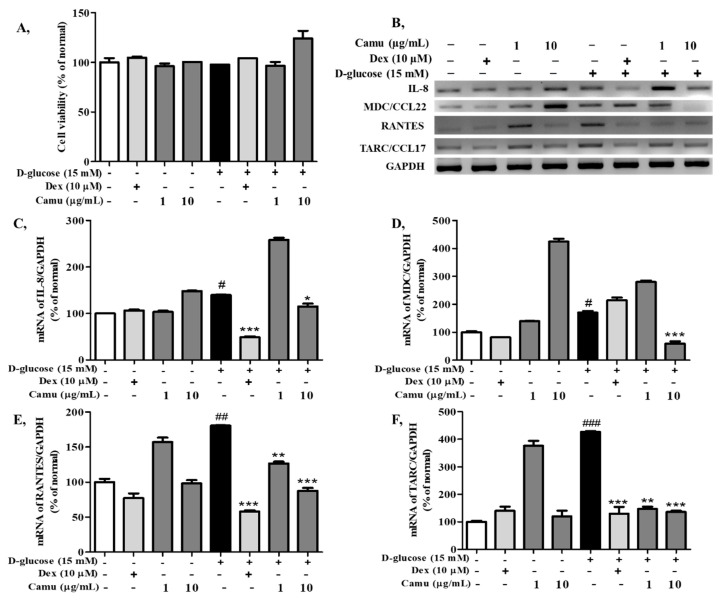
Effect of the camu-camu fruit extract on cell viability and mRNA expression of proinflammatory cytokines and chemokines in high glucose-induced HaCaT cells. Effect of the camu-camu fruit on the cell viability of high glucose-induced HaCaT cells (**A**). Expression of mRNA of IL-8, MDC, RANTES, and TARC in HaCaT cells under a high glucose-treated condition was measured by RT-PCR analysis (**B**). Band intensities of IL-8 (**C**), MDC (**D**), RANTES (**E**), and TARC (**F**) were calculated by densitometry and compared to GAPDH. The data are shown as the means ± SD of three independent experiments. # Significant differences between the untreated group and the high glucose-induced group (# *p* < 0.05; ## *p* < 0.01; ### *p* < 0.001). * Significant differences between the high glucose-induced group and the other groups (* *p* < 0.05; ** *p* < 0.01; *** *p* < 0.001).

**Figure 4 molecules-26-03174-f004:**
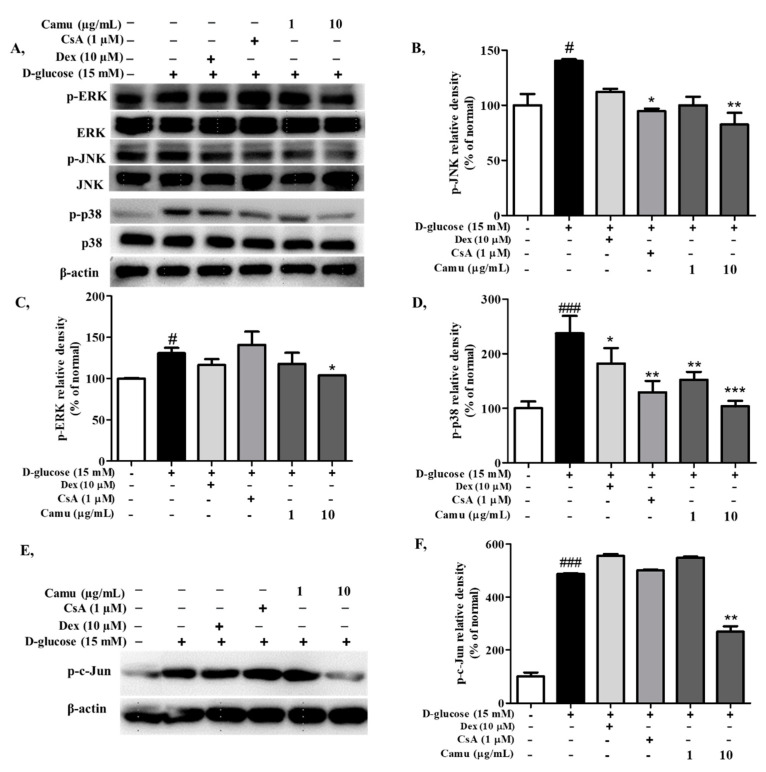
Effects of the camu-camu fruit on the MAPK/AP-1 signaling pathways in high glucose-stimulated HaCaT cells. JNK, p38, ERK, and their phosphorylation were analyzed by Western blotting (**A**). Phosphorylation of c-Jun was assessed by the Western blot analysis (**E**). Band intensities for p-JNK (**B**), p-ERK (**C**), p-p38 (**D**), and p-c-Jun (**F**) were measured by densitometry and normalized; the percentage was calculated on the basis of the level of β-actin. The data are shown as the means ± SD of three independent experiments. # Significant differences between the untreated group and the high glucose-induced group (# *p* < 0.05; ### *p* < 0.001). * Significant differences between the high glucose-induced group and the other groups (* *p* < 0.05; ** *p* < 0.01; *** *p* < 0.001).

**Figure 5 molecules-26-03174-f005:**
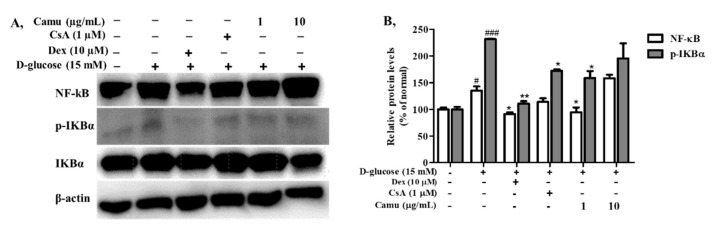
Effects of the camu-camu fruit on the NF-κB signaling pathway in high glucose-stimulated HaCaT cells. (**A**) The expression of NF-κB, IKBα and phosphorylation of IKBα were performed by Western blotting. Band intensities of NF-κB and p-IKBα (**B**) were measured by densitometry and normalized; the percentage was calculated on the basis of the level of β-actin. Values are shown as the means ± SD of three independent experiments. # Significant differences between the untreated group and the high glucose-induced group (# *p* < 0.05; ### *p* < 0.001). * Significant differences between the high glucose-induced group and the other groups (* *p* < 0.05; ** *p* < 0.01).

**Figure 6 molecules-26-03174-f006:**
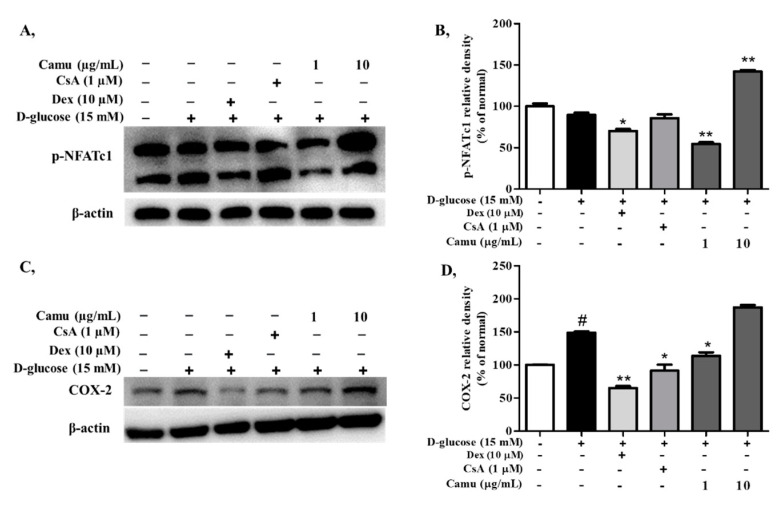
Effects of the camu-camu fruit on the NFATc1 and COX-2 signaling pathways in high glucose-stimulated HaCaT cells. Phosphorylation of NFATc1 (**A**) and COX-2 (**C**) was evaluated by Western blot analysis. Band intensities for p-NFATc1 (**B**) and COX-2 (**D**) were quantified using densitometry and normalized; the percentage was calculated on the basis of the level of β-actin. The data are shown as the means ± SD of three independent experiments. # Significant differences between the untreated group and the high glucose-induced group (# *p* < 0.05). * Significant differences between the high glucose-induced group and the other groups (* *p* < 0.05; ** *p* < 0.01).

**Figure 7 molecules-26-03174-f007:**
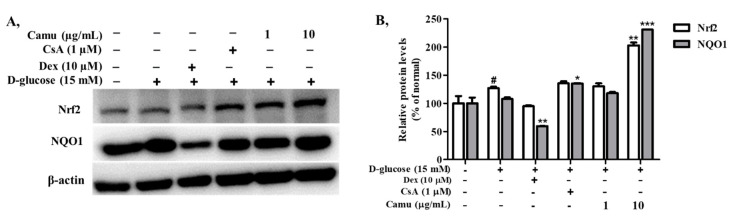
Regulatory effects of the camu-camu fruit extract on the Nrf2/ARE pathway in high glucose-stimulated HaCaT cells were assessed by Western blot analysis (**A**) and band intensities for Nrf2 and NQO-1 were quantified by densitometry and normalized; the percentage was calculated on the basis of the level of β-actin (**B**). The data are shown as the means ± SD of at least two independent experiments. # Significant differences between the untreated group and the high glucose-induced group (# *p* < 0.05) * Significant differences between the high glucose-induced group and the other groups (* *p* < 0.05; ** *p* < 0.01; *** *p* < 0.001).

**Figure 8 molecules-26-03174-f008:**
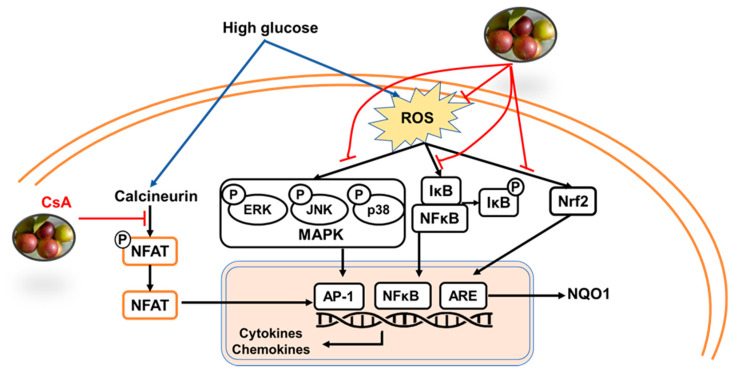
Anti-inflammatory effects of the camu-camu fruit on high-glucose-induced keratinocytes.

**Table 1 molecules-26-03174-t001:** Pharmacological effects of active compounds found in the camu-camu fruit.

Active Compound	Structure	Pharmacological Effect	Reference
	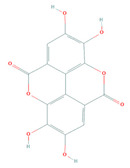	Anti-inflammatory	[25,26]
		Immunomodulatory	[27]
		Antiallergic	[28]
Ellagic acid (PubChem: 5281855)		Anticancer	[29]
		Antioxidative	[30]
		Antiaging	[31]
		Anti-diabetes	[32]
Quercetin (PubChem: 5280343)	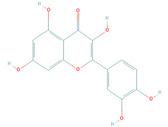	Immunomodulatory	[33]
		Anti-inflammatory	[34]
		Antioxidative	[35]
		Antiaging	[36]
		Anticancer	[37]

## Data Availability

The data presented in this study are available on request from the corresponding author.

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
