# Peer review of "Camu-Camu Fruit Extract Inhibits Oxidative Stress and Inflammatory Responses by Regulating NFAT and Nrf2 Signaling Pathways in High Glucose-Induced Human Keratinocytes"

_molecules, 2021, doi:10.3390/molecules26113174_

Round 1
Reviewer 1 Report
This manuscript reveals the mechanisms underlying oxidative stress reduction and anti-inflammatory effects of camu-camu fruit on high glucose-induced skin damage, showing that the camu-camu fruit modulated NF-κB/AP-1, MAPK, and NFAT signaling pathways related to inflammation by down regulating the expression of proinflammatory cytokines and chemokines. The topic is interesting and the experimental design was reasonable. I recommend its publication in Molecules after minor revision.
- A list of abbreviations accompanying their explanations should be provided.
- The references should be listed as the requirements of the Journal.
Author Response
Response to Reviewer 1 Comments
Point 1: A list of abbreviations accompanying their explanations should be provided.
Response 1: Thank you for your recommendation. A list of abbreviations accompanying their explanations was added in the revised manuscript. Please check Page 1,2 line 35-53.
Abbreviations:
AP-1: activator protein-1
ARE: antioxidant responsive element
DCFH-DA: 2’,7’-dichlorofluorescin diacetate
DPPH: 1,1-diphenyl-2-picrylhydrazyl
ERK: extracellular signal-regulated kinase
HaCaT: human skin keratinocyte cell line
HO-1: heme oxygenase 1
IL-8: interleukin 8
JNK: c-Jun N-terminal kinase
MAPK: mitogen-activated protein kinase
MDC: human macrophage-derived chemokine
NFAT: nuclear factor of activated T cells
NF-κB: nuclear factor-light-chain-enhancer of activated B cells
NQO1: NAD(P)H: quinone oxidoreductase1
Nrf2: nuclear factor E2-related factor 2
RANTES: regulated upon activation, normal T cell expressed, and secreted
ROS: reactive oxygen species
TARC: thymus- and activation-regulated chemokine
Point 2: The references should be listed as the requirements of the Journal.
Response 2: Thank you for your indication. The references were checked and corrected as the requirements of the Journal. Please check in the revised manuscript Page 13-16 line 425-542.
References
- Rufino, M. do, S.M.; Alves, R.E.; de Brito, E.S.; Pérez-Jiménez, J.; Saura-Calixto, F.; Mancini-Filho, J. Bioactive Compounds and Antioxidant Capacities of 18 Non-Traditional Tropical Fruits from Brazil. Food Chemistry 2010, 121, 996–1002.
- Castro, J.C.; Maddox, J.D.; Imán, S.A. Camu-camu— Myrciaria dubia (Kunth) McVaugh. In Exotic Fruits; Elsevier: Netherlands 2018; pp. 97–105.
- Langley, P.C.; Pergolizzi, J.V.; Taylor, R.; Ridgway, C. Antioxidant and Associated Capacities of Camu Camu (Myrciaria dubia): A Systematic Review. J Altern Complement Med. 2015, 21, 8–14.
- Azevêdo, J.C.S.; Borges, K.C.; Genovese, M.I.; Correia, R.T.P.; Vattem, D.A. Neuroprotective Effects of Dried Camu-Camu (Myrciaria dubia HBK McVaugh) Residue in C. Elegans. Food Research International 2015, 73, 135–141.
- Fujita, A.; Sarkar, D.; Wu, S.; Kennelly, E.; Shetty, K.; Genovese, M.I. Evaluation of Phenolic-Linked Bioactives of Camu-Camu (Myrciaria dubia Mc. Vaugh) for Antihyperglycemia, Antihypertension, Antimicrobial Properties and Cellular Rejuvenation. Food Research International 2015, 77, 194–203.
- Fidelis, M.; do Carmo, M.A.V.; da Cruz, T.M.; Azevedo, L.; Myoda, T.; Miranda Furtado, M.; Boscacci Marques, M.; Sant’Ana, A.S.; Inês Genovese, M.; Young Oh, W.; et al. Camu-Camu Seed (Myrciaria dubia) - From Side Stream to Anantioxidant, Antihyperglycemic, Antiproliferative, Antimicrobial, Antihemolytic, Anti-Inflammatory, and Antihypertensive Ingredient. Food Chem. 2020, 310, 125909.
- Da Silva, F.C.; Arruda, A.; Ledel, A.; Dauth, C.; Romão, N.F.; Viana, R.N.; de Barros Falcão Ferraz, A.; Picada, J.N.; Pereira, P. Antigenotoxic Effect of Acute, Subacute and Chronic Treatments with Amazonian Camu-Camu (Myrciaria dubia) Juice on Mice Blood Cells. Food Chem Toxicol. 2012, 50, 2275–2281.
- Chatterjee, S. Oxidative Stress, Inflammation, and Disease. In Oxidative Stress and Biomaterials; Elsevier: Netherlands, 2016; pp. 35–58.
- Lin, X.; Huang, T. Oxidative Stress in Psoriasis and Potential Therapeutic Use of Antioxidants. Free Radic Res. 2016, 50, 585–595.
- Ji, H.; Li, X.K. Oxidative Stress in Atopic Dermatitis. Oxid Med Cell Longev. 2016, 2016, 2721469.
- Umpierrez, G.E.; Hellman, R.; Korytkowski, M.T.; Kosiborod, M.; Maynard, G.A.; Montori, V.M.; Seley, J.J.; Van den Berghe, G.; Endocrine Society Management of Hyperglycemia in Hospitalized Patients in Non-Critical Care Setting: An Endocrine Society Clinical Practice Guideline. J Clin Endocrinol Metab. 2012, 97, 16–38.
- Mouri, M.; Badireddy, M. Hyperglycemia. In StatPearls; StatPearls Publishing: Treasure Island (Florida), United States 2020.
- Shah, G.N.; Morofuji, Y.; Banks, W.A.; Price, T.O. High Glucose-Induced Mitochondrial Respiration and Reactive Oxygen Species in Mouse Cerebral Pericytes Is Reversed by Pharmacological Inhibition of Mitochondrial Carbonic Anhydrases: Implications for Cerebral Microvascular Disease in Diabetes. Biochem Biophys Res Commun. 2013, 440, 354–358.
- Dangwal, S.; Stratmann, B.; Bang, C.; Lorenzen, J.M.; Kumarswamy, R.; Fiedler, J.; Falk, C.S.; Scholz, C.J.; Thum, T.; Tschoepe, D. Impairment of Wound Healing in Patients With Type 2 Diabetes Mellitus Influences Circulating MicroRNA Patterns via Inflammatory Cytokines. Arterioscler Thromb Vasc Biol. 2015, 35, 1480–1488.
- Yu, T.; Jhun, B.S.; Yoon, Y. High-Glucose Stimulation Increases Reactive Oxygen Species Production through the Calcium and Mitogen-Activated Protein Kinase-Mediated Activation of Mitochondrial Fission. Antioxid Redox Signal. 2011, 14, 425–437.
- Macian, F. NFAT Proteins: Key Regulators of T-Cell Development and Function. Nat Rev Immunol. 2005, 5, 472–484.
- Al-Daraji, W.I.; Grant, K.R.; Ryan, K.; Saxton, A.; Reynolds, N.J. Localization of Calcineurin/NFAT in Human Skin and Psoriasis and Inhibition of Calcineurin/NFAT Activation in Human Keratinocytes by Cyclosporin A. Journal of Investigative Dermatology 2002, 118, 779–788.
- Zhou, P.; Sun, L.J.; Dötsch, V.; Wagner, G.; Verdine, G.L. Solution Structure of the Core NFATC1/DNA Complex. Cell 1998, 92, 687–696.
- Khalaf, H.; Jass, J.; Olsson, P.-E. The Role of Calcium, NF-κB and NFAT in the Regulation of CXCL8 and IL-6 Expression in Jurkat T-Cells. Int J Biochem Mol Biol. 2013, 4, 150–156.
- Lee, S.I.; Yu, J.S. NFATc Mediates Lipopolysaccharide and Nicotine-Induced Expression of INOS and COX-2 in Human Periodontal Ligament Cells. Journal of dental hygiene science 2015, 15, 753–760.
- Ma, Q. Role of Nrf2 in Oxidative Stress and Toxicity. Annu Rev Pharmacol Toxicol. 2013, 53, 401–426.
- Ding, X.; Jian, T.; Wu, Y.; Zuo, Y.; Li, J.; Lv, H.; Ma, L.; Ren, B.; Zhao, L.; Li, W.; et al. Ellagic Acid Ameliorates Oxidative Stress and Insulin Resistance in High Glucose-Treated HepG2 Cells via MiR-223/Keap1-Nrf2 Pathway. Biomed Pharmacother. 2019, 110, 85–94.
- Liao, H.; Zhang, N.; Meng, Y.; Feng, H.; Yang, J.; Li, W.; Chen, S.; Wu, H.; Deng, W.; Tang, Q. Myricetin Alleviates Pathological Cardiac Hypertrophy via TRAF6/TAK1/MAPK and Nrf2 Signaling Pathway. Oxidative Medicine and Cellular Longevity 2019, 2019, 1–14.
- Schäfer, M.; Werner, S. Nrf2—A Regulator of Keratinocyte Redox Signaling. Free Radical Biology and Medicine 2015, 88, 243–252.
- Aslan, A.; Gok, O.; Beyaz, S.; Ağca, C.A.; Erman, O.; Zerek, A. Ellagic Acid Prevents Kidney Injury and Oxidative Damage via Regulation of Nrf-2/NF-κB Signaling in Carbon Tetrachloride Induced Rats. Mol Biol Rep. 2020, 47, 7959–7970.
- Rosillo, M.A.; Sánchez-Hidalgo, M.; Cárdeno, A.; Aparicio-Soto, M.; Sánchez-Fidalgo, S.; Villegas, I.; de la Lastra, C.A. Dietary Supplementation of an Ellagic Acid-Enriched Pomegranate Extract Attenuates Chronic Colonic Inflammation in Rats. Pharmacol Res. 2012, 66, 235–242.
- Seung, N.K.; Kim, Y.J.; Kim, T.S.; Sohn, E.H. Study on the Immunomodulatory Effects of Ellagic Acid and Their Mechanisms Related to Toll-like Receptor 4 in Macrophages. Korean Journal of Plant Resources 2012, 25, 561–567.
- Choi, Y.H.; Yan, G.H. Ellagic Acid Attenuates Immunoglobulin E-Mediated Allergic Response in Mast Cells. Biol Pharm Bull. 2009, 32, 1118–1121.
- Umesalma, S.; Sudhandiran, G. Ellagic Acid Prevents Rat Colon Carcinogenesis Induced by 1, 2 Dimethyl Hydrazine through Inhibition of AKT-Phosphoinositide-3 Kinase Pathway. Eur J Pharmacol. 2011, 660, 249–258.
- Devipriya, N.; Sudheer, A.R.; Menon, V.P. Dose-Response Effect of Ellagic Acid on Circulatory Antioxidants and Lipids during Alcohol-Induced Toxicity in Experimental Rats. Fundam Clin Pharmacol. 2007, 21, 621–630.
- Bae, J.Y.; Choi, J.S.; Kang, S.W.; Lee, Y.J.; Park, J.; Kang, Y.H. Dietary Compound Ellagic Acid Alleviates Skin Wrinkle and Inflammation Induced by UV-B Irradiation. Exp Dermatol. 2010, 19, 182-190.
- Rozentsvit, A.; Vinokur, K.; Samuel, S.; Li, Y.; Gerdes, A.M.; Carrillo-Sepulveda, M.A. Ellagic Acid Reduces High Glucose-Induced Vascular Oxidative Stress Through ERK1/2/NOX4 Signaling Pathway. Cell Physiol Biochem. 2017, 44, 1174–1187.
- Mendes, L.F.; Gaspar, V.M.; Conde, T.A.; Mano, J.F.; Duarte, I.F. Flavonoid-Mediated Immunomodulation of Human Macrophages Involves Key Metabolites and Metabolic Pathways. Sci Rep. 2019, 9, 14906.
- Beken, B.; Serttas, R.; Yazicioglu, M.; Turkekul, K.; Erdogan, S. Quercetin Improves Inflammation, Oxidative Stress, and Impaired Wound Healing in Atopic Dermatitis Model of Human Keratinocytes. Pediatric Allergy, Immunology, and Pulmonology 2020, 33, 69–79.
- Xu, D.; Hu, M.J.; Wang, Y.Q.; Cui, Y.L. Antioxidant Activities of Quercetin and Its Complexes for Medicinal Application. Molecules 2019, 24.
- Shin, E.J.; Lee, J.S.; Hong, S.; Lim, T.G.; Byun, S. Quercetin Directly Targets JAK2 and PKCδ and Prevents UV-Induced Photoaging in Human Skin. Int J Mol Sci. 2019, 20.
- Lee, T.J.; Kim, O.H.; Kim, Y.H.; Lim, J.H.; Kim, S.; Park, J.W.; Kwon, T.K. Quercetin Arrests G2/M Phase and Induces Caspase-Dependent Cell Death in U937 Cells. Cancer Letters 2006, 240, 234–242.
- Liu, T.; Zhang, L.; Joo, D.; Sun, S.C. NF-κB Signaling in Inflammation. Signal Transduct Target Ther. 2017, 2.
- Oeckinghaus, A.; Ghosh, S. The NF-KappaB Family of Transcription Factors and Its Regulation. Cold Spring Harb Perspect Biol. 2009, 1, a000034.
- Flockhart, R.J.; Diffey, B.L.; Farr, P.M.; Lloyd, J.; Reynolds, N.J. NFAT Regulates Induction of COX-2 and Apoptosis of Keratinocytes in Response to Ultraviolet Radiation Exposure. FASEB J. 2008, 22, 4218–4227.
- Jaime, P.L.A.; Francisca, das C. do A.S. Camu-Camu Super Fruit (Myrciaria dubia (H.B.K) Mc Vaugh) at Different Maturity Stages. Afr. J. Agric. Res. 2016, 11, 2519–2523.
- Furukawa, M.; Yamada, K.; Kurosawa, M.; Shikama, Y.; Wang, J.; Watanabe, M.; Kanekura, T.; Matsushita, K. High Concentration of Glucose Induces Filaggrin-1 Expression through AP-1 in Skin Keratinocytes. Journal of Dermatological Science 2020, 98, 137–140.
- Wang, Y.; Zhang, J.; Zhang, L.; Gao, P.; Wu, X. Adiponectin Attenuates High Glucose-Induced Apoptosis through the AMPK/P38 MAPK Signaling Pathway in NRK-52E Cells. PLoS ONE 2017, 12, e0178215.
- Atsaves, V.; Leventaki, V.; Rassidakis, G.Z.; Claret, F.X. AP-1 Transcription Factors as Regulators of Immune Responses in Cancer. Cancers (Basel) 2019, 11.
- Asadi, F.; Razmi, A.; Dehpour, A.R.; Shafiei, M. Tropisetron Inhibits High Glucose-Induced Calcineurin/NFAT Hypertrophic Pathway in H9c2 Myocardial Cells. J Pharm Pharmacol. 2016, 68, 485–493.
- Garcia-Vaz, E.; McNeilly, A.D.; Berglund, L.M.; Ahmad, A.; Gallagher, J.R.; Dutius Andersson, A.M.; McCrimmon, R.J.; Zetterqvist, A.V.; Gomez, M.F.; Khan, F. Inhibition of NFAT Signaling Restores Microvascular Endothelial Function in Diabetic Mice. Diabetes 2020, 69, 424–435.
- Martínez-Martínez, S.; Redondo, J.M. Inhibitors of the Calcineurin/NFAT Pathway. Curr Med Chem. 2004, 11, 997–1007.
- Lee, S.S.; Tan, A.W.H.; Giam, Y.C. Cyclosporin in the Treatment of Severe Atopic Dermatitis: A Retrospective Study. Ann Acad Med Singap. 2004, 33, 311–313.
- Polce, S.A.; Burke, C.; França, L.M.; Kramer, B.; de Andrade Paes, A.M.; Carrillo-Sepulveda, M.A. Ellagic Acid Alleviates Hepatic Oxidative Stress and Insulin Resistance in Diabetic Female Rats. Nutrients 2018, 10.
- Ngo, H.T.T.; Hwang, E.; Kang, H.; Park, B.; Seo, S.A.; Yi, T.H. Anti-Inflammatory Effects of Achillea Millefolium on Atopic Dermatitis-Like Skin Lesions in NC/Nga Mice. Am J Chin Med. 2020, 48, 1121-1140.
Reviewer 2 Report
The authors in their article present research related to the anti-oxidative and anti-inflammatory effects of an extract derived from the fruit of Myrciaria dubia (camu-camu). These effects were examined toward oxidative stress-induced, high glucose treated HaCaT cells. The extract was found to exert its activity through modulation of the NF-κB/AP-1, MAPK, and NFAT signaling pathways downregulating proinflammatory cytokines and chemokines. Additionally, the upregulation of Nrf2 and NQO1 were induced by camu-camu.
Overall, the authors present interesting findings adding to the current knowledge regarding the bioactivity of Myrciaria dubia. The manuscript is well written and research clearly presented. I have the following questions for the authors regarding their study:
The increase in fluorescence intensity between the control and glucose-treated cells in the histograms does not seem very significant. Could the authors explain the performed analysis of the fluorescence intensity. Did the authors consider different treatment periods with glucose to achieve a more pronounced increase in ROS generation.
In Figure 2B and 2C please add information regarding the probe used to determine ROS generation, either in the figure description or to the axis of the graph or histograms.
In Figure 5 in the determination of p-IKBα levels with Western blot, the changes in the levels of p-IKBα between samples are undistinguishable. Do the authors have a better quality repetition of this experiment that they could include.
The authors propose the use of the camu-camu extract as an alternative to treatment with cyclosporin A. Since the constituent of camu-camu, ellagic acid, has been demonstrated to exert anti-inflammatory and anti-oxidant activity, could the authors comment on the applicability of the cam-camu extract in comparison to ellagic acid in skin inflammation treatment.
I would suggest adding a graphical scheme of the obtained results at the end of the manuscript.
Round 2
Reviewer 2 Report
The authors have thoroughly addressed my questions and I therefore find the manuscript appropriate for publication.